# Storage Time and Temperature on the Sensory Properties Broccoli

**DOI:** 10.3390/foods8050162

**Published:** 2019-05-12

**Authors:** Robert Pellegrino, Jennifer Wheeler, Carl E. Sams, Curtis R. Luckett

**Affiliations:** 1Department of Food Science, Institute of Agriculture, University of Tennessee, Knoxville, TN 37996, USA; pellegrino.robert@gmail.com; 2Department of Plant Science, Institute of Agriculture, University of Tennessee, Knoxville, TN 37996, USA; jwheel22@utk.edu (J.W.); carlsams@utk.edu (C.E.S.)

**Keywords:** consumers, sensory evaluation, plant, broccoli, shelf-life

## Abstract

Typically, broccoli arrives at the store within 7–14 days of harvest and is kept refrigerated until purchased or considered waste. To date, information has been limited on how this time on the shelf or storage temperature affects the sensory attributes that contribute to broccoli purchase or repurchase. In this study, 100 consumers performed acceptance tests and a check-all-that-apply (CATA) section to characterize sensory changes in two cultivars of broccoli (‘Diplomat’ and ‘Emerald Crown’) stored at two temperatures (0 °C and 4 °C) over five time points: 0, 14, 21, 28, and 42 days. Due to quality degradation during storage, the overall liking of broccoli decreased regardless of holding temperature and variety. This was in accordance with a decrease in sweetness and an increase in bitterness intensity. However, there were differences between varieties in which Diplomat had more sensory changes at higher temperatures and only Emerald showed negative changes to its appearance in color. Lastly, the CATA data revealed the attributes responsible for modulating the consumer acceptance of broccoli such as tastes, colors and flavors (e.g., grassy, musty, dirt-like). This information can be used to better inform shelf-life determinations of broccoli. Additionally, these changes in taste, odor, texture, and color can inform those interested in investigating the biochemical processes related to broccoli storage.

## 1. Introduction

The United States is the third largest producer of broccoli, with California producing a majority of the US production (90%). For instance, in 2010, 121,700 acres of broccoli was harvested with an estimated value of 648 billion dollars (USD) [1]. In the US, the average consumption, according to a 2017 consumer report, was 7.1 pounds per capita [2]. Commercially, broccoli is chilled to 0 °C within five hours and packed on ice for transport [3]. The broccoli arrives at the store within 7 to 14 days of harvest and is placed in a refrigerated grocery aisle until purchased or considered waste (typically around one week) [3,4]. As harvested broccoli ages, it senesces with time causing the nutritional and sensory profiles to change. Broccoli has a rich content of bioactive molecules such as vitamin C, glucosinolates, phenolic compounds, and carotenoids [5]; however, it is a highly perishable product and its nutritional, visual, and flavor profiles depend on its storage conditions. A timely harvest, cooling, controlled atmosphere, packaging and the use of less perishable cultivars are some of the tools used to extend the shelf-life of broccoli [6,7]. Many of these practices control moisture levels, reduce the rate of respiration, slow down enzymatic reactions, and decrease the production and action of ethylene. Ethylene is well known to be involved in the maturation and senescence of fruits and vegetables [8]. Several studies have extensively investigated the overall appearance and nutritional quality during post-harvest storage strategies across a large range of broccoli cultivars [4,9,10,11]. However, there is limited data on the changes in consumer perception driven by changing biochemical profiles of broccoli during storage. Moreover, research needs to consider techniques of preparation in which many consumers prepare their broccoli cooked rather than raw [12,13,14,15].

An established technique to gauge the consumer liking of a food product is through consumer acceptance testing [16]. Overall liking is a dominant predictor of customer acceptability, purchase decision, and product consumption; and brassica vegetables are no exception to this rule [11,17]. Beyond overall liking, other attribute categories such as appearance, flavor, and texture, which may be further divided into colors/shape, basic tastes, aromatic notes, and detailed texture descriptors [18]. This root-level of sensory attributes, which drives overall liking, has historically been ascertained by a trained panel of experts using conventional descriptive analysis (DA) [19], but recent research has shown similar data can be captured from consumers using rapid descriptive methods (RDM) [20,21]. One of the RDMs shown to be effective at mimicking the data from DA is check-all-that-apply (CATA), in which consumers simply select the attributes that are present in each sample. While the inclusion of CATA as an RDM is still questionable, since the intensity of the attributes is not assessed using CATA, this method has been shown several times to mirror the data from DA [20,21].

The current study was designed to characterize changes to aroma/flavor attributes for two broccoli cultivars (*Brassica oleracea* cvs ’Diplomat’ and ’Emerald Crown’) with storage conditions: 0 °C and 4 °C temperatures at 0, 14, 21, 28 and 42 days of storage. A lexicon of 24 appearance and flavor descriptors found from a range of common broccoli cultivars within the market was created. A consumer acceptance test with a CATA section was used to measure the effects on flavor profile and their impact on consumer liking and willingness to purchase. To date, steaming has shown to be the least damaging to nutrients while maintaining the acceptance of consumers; thus, this cooking technique was chosen for the current study [15]. We hypothesize sensory properties responsible for consumer liking are regulated by storage time and temperature. 

## 2. Materials and Methods

### 2.1. Broccoli Production

Broccoli was purchased from a local producer in East Tennessee who utilized standard production practices recommended in the Southeastern U.S. 2018 Vegetable Crop Handbook [22].

### 2.2. Broccoli Harvest and Storage

Two cultivars of broccoli, ‘Emerald Crown’ and ‘Diplomat’, were harvested when their head size diameter was 10 to 15 cm. Harvested broccoli was separated into two treatments: 1) iced and 2) non-iced. Iced treatments were placed in coolers in the field, immediately topped with an ice slurry made up of crushed ice that was partially melted in the field, and then transported to a cold storage room and maintained at 0 °C. Broccoli was not submerged in water during transport as water drained out the bottom of the coolers. Non-iced treatments were placed in waxed corrugated cardboard boxes for transport and then placed in a cold storage room maintained at 4 °C. Watch Dog^®^ data loggers (Spectrum^®^ Technologies, Inc., Aurora, IL, USA) were placed into one head of broccoli per storage container, and the temperature was recorded every 30 min for the duration of storage. Harvest lasted approximately two hours per cultivar with a transportation time of approximately 45 min to cold storage rooms from the field. Iced samples were maintained in iced coolers that were drained to prevent water accumulation for the duration of the study. Iced broccoli was cooled to 2.5 °C within 45 min of harvesting and placing into coolers in the field, while non-iced broccoli was cooled to 6 °C within six hours of harvesting and four and a half hours of placing in a cold storage room set at 4 °C. Subsamples of broccoli were removed from storage at 0, 14, 21, 28, and 42 days post-harvest for sensory analysis.

### 2.3. Sensory Analysis

One hundred untrained broccoli consumers (see Table 1) were recruited for each time point through an online database managed by the Center for Sensory Science at the University of Tennessee. All participants reported no impairment to vision, smell, or taste and participated in the test on a voluntary basis. Experimental protocol was conducted according to the Declaration of Helsinki for studies on human subjects and approved by the University of Tennessee IRB review for research involving human subjects (IRB #17-04056-XP).

Prior to testing, broccoli from each temperature condition (0 and 4 °C) and cultivar (‘Diplomat’ and ‘Emerald Crown’) was trimmed, and its florets (diameter 3–4 cm) were steamed for five minutes. Steam time and procedure was chosen from past research showing increased liking for medium-firm cooked broccoli at a similar steaming time [14,15]. Broccoli from each of these four conditions were served to participants at room temperature in off-white, noise-controlled sensory booths. A floret from each condition (4 total/time point) was placed on a circular white plate (10 cm diameter) labeled with a random three-digit code. The serving order was randomized across consumers following a Latin square design. Before tasting, consumers rated the appearance liking, green color intensity, and aroma liking/intensity. They were then instructed to eat at least half the sample and rate the liking for overall, flavor, texture, aftertaste and the intensity for flavor, sweetness, bitterness, and aftertaste. Liking questions used a 9-point hedonic scale (“Like extremely” to “Dislike extremely”), while intensity questions used a 15-cm line scale (“Extremely weak” to “Extremely strong”) with a 7.5 midpoint anchor. Lastly, they were asked to check all that apply (CATA) regarding attributes perceived in the sample (Table 2 and Table 3). This procedure was repeated across five days that corresponded with the storage time points starting with harvest (or Day 0). Consumer data was recorded with RedJade sensory software (RedJade Software Solutions, LLC, Redwood Shores, CA, USA, 2019).

### 2.4. Data Analysis

A three-way analysis of variance (ANOVA) was used to evaluate consumer overall liking and attribute scores (hedonic and intensity) with cultivar, storage temperature, storage time, and their interactions. All ANOVA factors were considered fixed and Student t-tests were used where appropriate. Differences between conditions with CATA data were estimated using the frequency that each CATA attribute was chosen and were assessed with Cochran’s Q Test and Cochran’s Armitage Trend Test [23,24]. Additionally, McNemar’s tests were done for CATA data between varietals at Day 0, collapsing temperature conditions. Bivariate correlations were analyzed with Pearson’s *r*. To assess how overall liking was modulated by attributes present in the samples, the CATA descriptor counts were used to predict overall liking. A partial least squares regression model was constructed using the average overall liking scores and CATA descriptor frequencies. All data were analyzed using JMP Pro 13.0.0 (SAS Institute, Cary, NC, USA).

## 3. Results

### 3.1. Consumer Acceptance

As shown in Figure 1, there was a decrease in overall liking with increasing storage time (F_4, 1980_ = 7.06, *p* < 0.001) regardless of broccoli cultivar (F_1, 1980_ = 2.35, *p* = 0.13) or storage temperature (F_1, 1980_ = 0.06, *p* = 0.80). The first significant drop in overall liking, compared to fresh, was at 28 or more days of storage. However, the broccoli cultivars responded differently to the temperature at which they were stored (F_1, 1980_ = 9.73, *p* = 0.002). ‘Diplomat’ was liked more when stored at lower temperatures (mean (M) = 6.45, standard deviation (SD) = 1.73) than at higher temperatures (M = 5.96, SD = 1.96; F_1, 998_ = 17.59, *p* < 0.001), while ‘Emerald Crown’ was unaffected by storage temperatures (M = 6.22, SD = 1.76; F_1, 998_ = 0.01, *p* = 0.93). The same effects and their interactions were seen for flavor liking, which was highly correlated with overall liking (*r* = 0.91). Texture liking showed a significant decrease with only the higher storage temperature (F_4, 995_ = 4.18, *p* = 0.002). Aftertaste and aroma liking were also positively correlated with overall liking (*r* = 0.71 and *r* = 0.53, respectively). Liking scores for other attributes are shown in Figure 2.

Concerning intensity attributes, the intensity of aroma (F_4, 1980_ = 6.03, *p* < 0.001), and sweetness (F_4, 1980_ = 21.08, *p* < 0.001) decreased with time; while bitterness increased (F_4, 1980_ = 5.98, *p* < 0.001), and aftertaste showed no changes (F_4, 1980_ = 0.82, *p* = 0.51. Aroma intensity followed a similar drop-off to overall liking (28+ days of storage) and an immediate decline in sweetness was observed while bitter changes were more stochastic (see Figure 3). However, it is important to note that bitterness had an observable (yet not statistically significant) interaction with higher storage temperatures affecting ‘Diplomat’ (mean difference (MD = 0.86)) more than ‘Emerald Crown’ (MD = 0.19; *p* = 0.053). Similarly, sweet intensity showed an interaction with cultivar and storage temperature (F_1, 1980_ = 4.70, *p* = 0.03), while flavor intensity had an interaction with storage time and temperature (F_4, 1980_ = 4.37, *p* = 0.002). Sweetness significantly declined during higher storage temperatures for ‘Diplomat’ (F_1, 998_ = 4.69, *p* = 0.03), but this effect was not present for ‘Emerald Crown’ (F_1, 998_ = 0.65, *p* = 0.42). Furthermore, flavor intensity decreased with time at lower holding temperatures (F_1, 995_ = 12.60, *p* < 0.001), while remaining stable with time at higher storage temperatures (F_4, 995_ = 1.33, *p* = 0.26). No other conditions or interactions for intensity attributes were significant (*p* > 0.05).

Due to several differences observed in attribute acceptances, CATA data from varietals were analyzed separately (see Table 2 and Table 3); however, no CATA differences on harvest day were found between varietals. Bitter and sweetness terms increased and decreased, respectively, during at 4 °C storage temperature for both cultivars; however, more significant trends were shown in ‘Diplomat’. Aftertaste intensity significantly increased over time for each cultivar. ‘Emerald Crown’ had a significant increase in yellow and decrease in green color term count at both storage temperatures, indicating color distortion with age, while no color changes were significant with ‘Diplomat’. 

### 3.2. Drivers of Liking using Partial Least Squares (PLS)

The CATA descriptor counts were able to predict overall liking scores relatively well, accounting for 78.7% of the variance in liking. In all, 10 CATA descriptors were found to be important (Variable Importance in Projection (VIP) >0.8) in predicting liking scores (Figure 4). VIP is an assessment of the importance of each variable in the creation of a PLS model. Additionally, Figure 4 displays the regression coefficients on the *x*-axis. Regression coefficients indicate the direction, while the VIP (*y*-axis) indicates the magnitude, a variable has on predicting overall liking. The most influential positive drivers of liking include sweet, green, and intense color. The most influential negative drivers of liking include off-flavor, musty, dirt-like, and bitter. 

## 4. Discussion

The current study reports on organoleptic changes to varieties within the *Brassica* genus during storage. Here, we show that several sensory changes occur during the storage of broccoli with some, but not all, changes being dependent on cultivar or holding temperatures, which in turn were sometimes dependent on each other. 

Overall, the liking of the broccoli samples decreased over time, independent of cultivar and holding temperatures. The largest drop in overall liking was observed at 28 days of storage. In other words, cultivar and hold temperatures did not prevent consumers from noticing a decrease in quality of the broccoli. In a retail environment, a grocer typically receives broccoli within 7–14 days post-harvest and sells this broccoli within the next week [3]. Our results demonstrate that the receiving or selling time of broccoli may have implications on overall liking of broccoli when eaten by the consumer. Therefore, producers and grocers should work to minimize distribution and shelf-storage times. Furthermore, culinary use of broccoli needs to be considered when determining the sensory shelf-life of the product. More specifically, broccoli is typically deemed to be of low quality by many in the produce industry much earlier than this study shows [25,26,27]. While it was found that broccoli acceptability decreases over time, the effects of extended storage were not as severe as previously reported [25,26,27]. This finding suggests growers and shippers of broccoli should readdress their current views of storage time to reflect common cooking methods such as steaming. In the case of steamed broccoli, the produce does not show the rapid declines in quality often used to drive industry norms.

Independent of storage time, ‘Diplomat’ benefitted from lower storage temperature and was more susceptible to organoleptic changes at higher temperatures than the ‘Emerald Crown’. Inferring from the specific attribute changes, we can deduce several reasons for changes in overall liking. Sweetness and bitterness are both important taste attributes to the acceptance of a food product with an increase and decrease increasing acceptance, respectively [28,29]. An increase in storage time decreased overall liking and subsequently decreased the sweetness and increased the bitterness of the broccoli samples. Additionally, the texture liking of the broccoli dropped over time for the higher stored temperature. This may be due to a reduction in water loss with rapid cooling [30]; however, weight of the samples were not collected. Similar results were obtained from a 2004 study in which storage degradation of broccoli under different packaging conditions (at two temperatures, 4 and 10 °C) resulted in reduced texture quality. In their study, they showed, with instrumental texture measurements, the broccoli became softer across four weeks for the most common packaging material, polyvinyl chloride (PVC), at 4 °C [7]. However, ‘Diplomat’, unlike the ‘Emerald Crown’, decreased in sweetness in both intensity scores and descriptive CATA analysis. Furthermore, “Dirt-like”, a typically negative term, was increased only in the higher storage temperature for ‘Diplomat’, while color differences were seen in ‘Emerald Crown’. Differences among cultivars are not uncommon [11]. A study looking at several cultivars of broccoli (*n* = 5) and cauliflower (*n* = 4) used descriptive and consumer data (*n* = 100) to show taste, appearance, and overall liking differences. Consumers preferred samples with lower levels of bitter tasting glucosinolates (alkenyl and indole glucosinolates) and a higher sucrose content (mostly fructose and glucose). Most glucosinolates decline with storage, especially at higher temperatures [31], but some have been shown to increase into deterioration (4-methoxglucobrassicin) [32]. Thus, maybe the decrease of sweetness unmasked the underlying bitterness [33] or perhaps other factors were at play. For instance, there is a high variability in the sensitivity to bitterness across the population due to genetics, which may be reflected in the variability of bitterness intensity across time points observed in our study (each of which had a different set of participants) [34]. Additionally, a large variation across broccoli samples existed for flavor terms such as green/ grassy, spicy, cabbage-like, and leek-like [11].

There were increasing levels of aftertaste revealed by the CATA data collected. Similarly, aftertaste and dirt-like were shown to be important variables to predict overall liking. CATA also showed a clear effect of storage temperature with higher temperatures. 

## 5. Conclusions

Similar to most fresh produce, broccoli is a nutritious but perishable product whose purchase by consumers depends on its sensory attributes. Once harvested, broccoli continues to respire and senesce; and its quality decreases, reducing willingness to purchase. In particular, storage time imparts negative changes to appearance, basic tastes, flavor attributes, and texture. However, many of these changes can be slowed or deterred by decreasing the storage temperature and choosing a more resilient cultivar. For example, taste (bitter and sweet) and texture are major determinants of broccoli liking for both cultivars. Specifically, the Diplomat variety was more susceptible to negative taste outcomes and flavor changes (e.g. increase in attributes like cabbage and dirt-like) while the color degraded more for Emerald. Therefore, practitioners should use this information to implement procedures that could minimize harvest-to-consumption time and decrease storage temperatures while testing additional broccoli cultivars to determine quality resilience to these factors.

## Figures and Tables

**Figure 1 foods-08-00162-f001:**
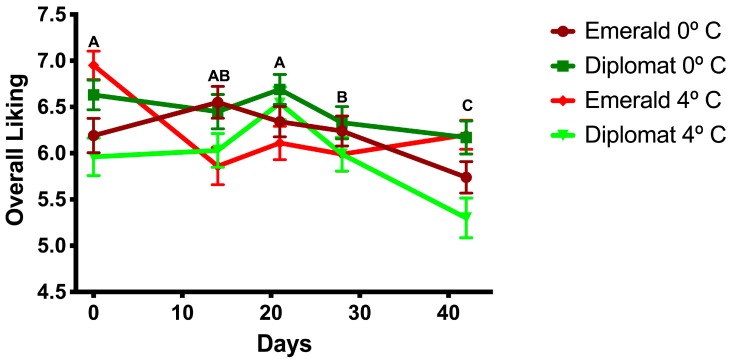
Consumer acceptance of broccoli cultivars over time depending on storage temperatures. Means of time points are compared and differences are represented with ascending letters.

**Figure 2 foods-08-00162-f002:**
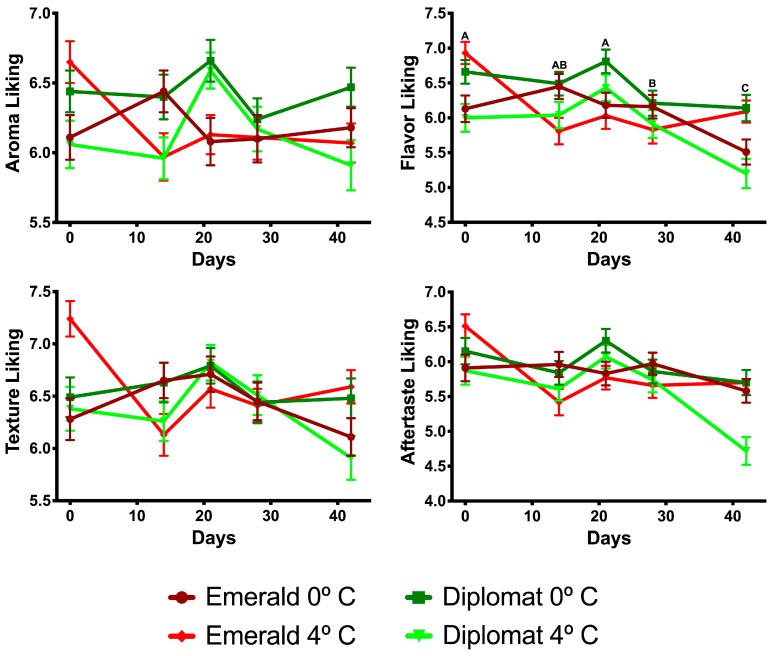
Consumer liking scores of the attributes of broccoli cultivars over time depending on storage temperatures. Means of time points are compared in models without significant interaction terms and differences are represented with ascending letters.

**Figure 3 foods-08-00162-f003:**
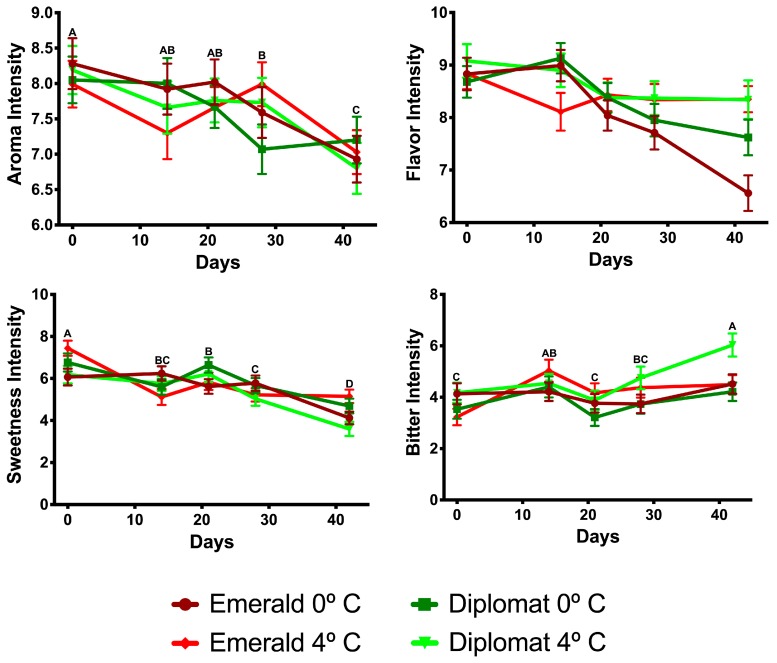
Sensory attribute intensities of broccoli cultivars over time depending on storage temperatures. Means of time points are compared in models without significant interaction terms and differences are represented with ascending letters.

**Figure 4 foods-08-00162-f004:**
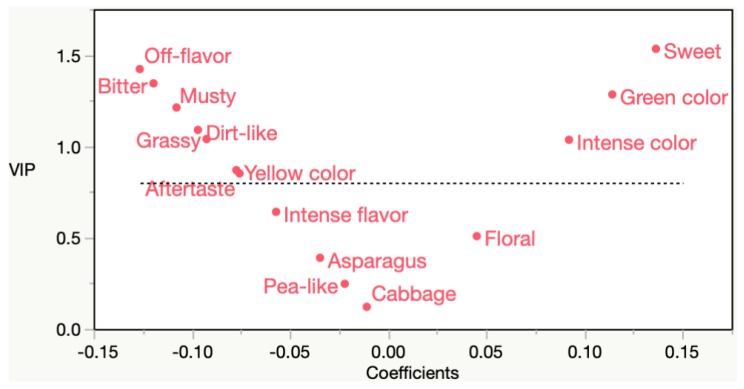
Variable Importance in Projection (VIP) and βcoefficient for Check-All-That-Apply (CATA) frequencies predicting overall liking.

**Table 1 foods-08-00162-t001:** Demographics of participants for each testing time point.

	Time Points in Days				
	Day 0	Day 14	Day 21	Day 28	Day 42
**Sex (*n*)**					
Men	30	26	38	29	28
Women	70	74	62	71	72
**Age Range (%)**					
18–24	30%	28%	19%	17%	21%
25–34	36%	33%	39%	49%	45%
35–44	11%	14%	12%	14%	13%
45–54	10%	10%	18%	11%	13%
55–64	13%	15%	12%	9%	7%
65 and Over	0%	0%	0%	0%	1%
**Ethnicity (%)**					
White/Caucasian	81%	76%	75%	75%	78%
Hispanic/Latino	3%	2%	5%	1%	3%
Black/African American	6%	8%	2%	4%	6%
Native American	0%	0%	0%	0%	0%
Asian/Pacific Islander	8%	12%	17%	18%	12%
Other	2%	2%	1%	2%	1%

**Table 2 foods-08-00162-t002:** Contingency table showing the frequency of each Check-All-That-Apply (CATA) attribute mentioned for each storage time point depending on temperature condition (0 °C or 4 °C) for the ‘Diplomat’ broccoli cultivar. The last two columns report the test statistic for Cochran’s Q test (Q) and Cochran Armitage Trend test (Z).

Attribute	‘Diplomat’													
	0 Days		14 Days		21 Days		28 Days		42 Days		Q		Z	
	0 °C	4 °C	0 °C	4 °C	0 °C	4 °C	0 °C	4 °C	0 °C	4 °C	0 °C	4 °C	0 °C	4 °C
Dirt-like	10	10 ^b^	12	9 ^b^	6	11 ^b^	11	11 ^b^	11	27 ^a^	2.42	20.29 ***	0.15	3.46 ***
Aftertaste	46	40 ^b^	54	59 ^ab^	46	49 ^ab^	40	54 ^ab^	48	68 ^a^	3.88	18.34 **	−0.36	3.54 ***
Sweet	41	38 ^a^	39	35 ^a^	50	41 ^a^	36	31 ^ab^	33	16 ^b^	7.01	17.17 **	−1.23	−3.35 **
Bitter	21	27	27	30	18	24	12a	27	28 a	43	10.10 *	10.32 *	0.33	2.19 *
Cabbage	22	22	41	33	35	36	33	42	29	33	9.13	9.37	0.62	1.99 *
Off-flavor	11	21	13	18	14	16	19	15	22	30	5.91	8.90	2.39*	1.34
Yellow Color	14	15	21	17	19	16	18	14	18	26	1.70	6.34	0.52	1.76
Musty	9	17	17	20	15	15	14	13	17	25	3.43	6.03	1.34	0.99
Green Color	81	76	80	73	79	71	79	76	75	63	1.27	5.83	−1.04	−1.79
Grassy	34	44	35	46	37	39	40	36	38	49	0.97	4.86	0.79	0.25
Floral	18	14	11	9	10	7	10	9	16	8	4.96	3.61	-0.47	−1.38
Pea-like	14	11	12	15	10	13	18	12	11	15	3.33	1.08	-0.20	0.59
Asparagus	9	10	15	12	10	12	16	12	14	13	3.29	0.48	1.07	0.62
Minty	3	1	1	3	2	1	2	3	0	4	-	-	-	-
Piney	1	7	3	2	2	1	7	8	3	6	-	-	-	-
Putrid	1	4	5	5	1	5	3	7	5	7	-	-	-	-
Cucumber	4	7	6	3	6	7	10	11	6	10	-	-	-	-
Sour	6	3	5	7	2	5	3	6	3	9	-	-	-	-
Rotten Eggs	2	3	8	4	3	0	1	1	2	3	-	-	-	-
Freezer Burn	3	3	4	4	2	3	3	4	7	12	-	-	-	-

Cochran’s Q test observed value (Q) and Cochran Armitage Trent Test observed value (Z) represent the last two columns. “-“ represents not enough observations. * *p* < 0.05, ** *p* < 0.01, *** *p* < 0.001; Letters represent significant differences (*p* < 0.05) across days within temperature conditions. Attributes are ordered in descending Q value.

**Table 3 foods-08-00162-t003:** Contingency table showing the frequency of each Check-All-That-Apply (CATA) attribute mentioned for each storage time point depending on temperature condition (0 or 4 °C) for ‘Emerald Crown’. The last two columns report the test statistic for Cochran’s Q test (Q) and Cochran Armitage Trend test (Z).

Attribute	‘Emerald Crown’													
	0 Days		14 Days		21 Days		28 Days		42 Days		Q		Z	
	0 °C	4 °C	0 °C	4 °C	0 °C	4 °C	0 °C	4 °C	0 °C	4 °C	0 °C	4 °C	0 °C	4 °C
Aftertaste	45	35 ^b^	56	58 ^a^	50	41 ^ab^	38	54 ^ab^	44	57 ^a^	7.51	17.13 **	−0.94	2.77 **
Off-flavor	20	7 ^b^	12	27 ^a^	16	23 ^a^	18	16 ^ab^	19	19 ^ab^	2.92	15.41 **	0.18	1.44
Yellow Color	12	9 ^b^	21	27 ^a^	23	26 ^a^	28	19 ^ab^	29	25 ^a^	10.52 *	13.89 **	3.10 **	2.19 *
Bitter	29	16 ^b^	26	34 ^ab^	29	26 ^ab^	22	30 ^ab^	34	35 ^a^	4.19	12.08 *	0.55	2.63 **
Grassy	37	34	44	47	43	42	41	35	47	50	2.31	8.31	1.22	1.63
Sweet	36	52	39	35	40	37	38	38	29	36	3.26	8.12	−1.02	−2.06 *
Musty	12	10	18	19	12	18	15	19	20	22	3.97	5.76	1.30	2.11 *
Green Color	77	84	75	76	74	75	73	75	67	71	2.99	5.25	−1.62	−2.10 *
Dirt-like	5	7	7	11	4	9	10	5	18	6	-	3.41	-	0.76
Cabbage	23	24	37	32	38	30	36	35	31	32	6.97	3.20	1.09	1.31
Floral	9	16	12	11	4	8	14	11	11	12	-	3.16	-	0.84
Asparagus	10	7	9	11	16	11	14	7	13	8	3.05	2.05	0.95	−0.08
Pea-like	22	11	21	15	14	12	11	17	15	14	6.24	1.95	−1.86	0.71
Minty	2	3	2	1	2	1	4	2	2	1	-	-	-	-
Piney	4	4	1	5	2	5	3	4	6	4	-	-	-	-
Putrid	3	1	3	6	4	5	4	5	1	2	-	-	-	-
Cucumber	4	7	8	7	5	5	5	4	8	9	-	-	-	-
Sour	6	4	5	6	4	3	3	8	3	4	-	-	-	-
Rotten Eggs	2	1	3	4	2	0	3	3	3	2	-	-	-	-
Freezer Burn	2	2	5	10	2	4	3	5	5	3	-	-	-	-

Cochran’s Q test observed value (Q) and Cochran Armitage Trent Test observed value (Z) represent the last two columns. “-“ represents not enough observations. *p* < 0.05, ** *p* < 0.01, *** *p* < 0.001; Letters represent significant differences (*p* < 0.05) across days within temperature conditions. Attributes are ordered in descending Q value.

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
