# Peer review of "Storage Time and Temperature on the Sensory Properties Broccoli"

_foods, 2019, doi:10.3390/foods8050162_

Reviewer 1 Report

Manuscript foods-477679 Storage time and temperature on the sensory properties Broccoli

I believe that the out come of this research can contribute truely to broccoli producers in terms of strategically marketing their broccoli products to grocery stores. I would like to add one comment regarding the data analysis in that the direct comparison of Emerald crow  and Diplomat for CATA data may adds some insights especially  few terms that overlaps (i.e. dirt like, cabbage, off-flavor, yellow color etc),

Author Response

I believe that the outcome of this research can contribute truly to broccoli producers in terms of strategically marketing their broccoli products to grocery stores. I would like to add one comment regarding the data analysis in that the direct comparison of Emerald crow and Diplomat for CATA data may add some insights especially few terms that overlaps (i.e. dirt like, cabbage, off-flavor, yellow color etc),

We appreciate the feedback from the reviewer and agree this additional information would have practical benefits. Therefore, we have added the analysis and results.  We have also attached the SPSS.htm output from our analysis. However, as you will see, there was no CATA differences between the varieties at time point zero.

Reviewer 2 Report

An interesting study with fresh broccoli – there is not enough studies on the sensory properties of fresh vegetables. There are quite a few major points that the authors must take care of before this manuscript can be published. A good reference that the authors should cite in their manuscript – Jacobsson, A., Nielsen, T. & Sjöholm, I. Effects of type of packaging material on shelf-life of fresh broccoli by means of changes in weight, colour and texture. Eur Food Res Technol 218: 157.

1.      Line 12 – in the text you have mentioned 7-14 days

2.      Line change ‘organoleptic’ to ‘sensory’ in the manuscript

3.      Line 17 – why 42 days? That’s too long in my opinion since you say in the text that they sold by a weeks’ time after arrival. Day 7 should have been a timepoint.

4.      Line 32 – mention the usual shelf life in the grocery stores

5.      Why did you choose Broccoli? Give the rationale and some consumption statistics related to fresh broccoli

6.      Lines 52-54 – CATA is not a hybrid rapid descriptive method. The comparison is incorrect. CATA collects frequency data and does not indicate the intensities of the descriptors. Methods such as flash profiling and napping are rapid descriptive methods. So please change the explanation here.  

7.      Line 63 – this hypothesis is not correct – the hypothesis leads to your research objective – is that the case here?

8.      Line 78 – change ‘minutes’ to ‘min’

9.      Line 82 – explain 7/8 cooling

10.   Table 1 – why wasn’t the same consumer panel used throughout the study?

11.   Line 105 – how can it be a 15-point line scale? The word ‘point’ refers to a category scale. That is incorrect – line scales are used for trained panelists where the scores are measure using a ruler. I know that a data collection software was used which probably shows and record the number when the cursor is placed – but still it is not the right scale for untrained consumers. You should have used a 9-point category scale. Easier to understand because it corresponds to the 9-point hedonic scale in terms of the number of categories.

12.   Line 109 – version?

13.   Line 111 – random effects and fixed effects in the ANOVA model? Post-hoc mean separation technique?

14.   Line 118 – more explanation is required on the PLS. which stat package was used for the analyses?

15.   Line 123 – be consistent when you write ANOVA results as F values. Check throughout the results section.

16.   Line 131 – an r of 0.53 is not that high.

17.   Line 139 – the aftertaste results does not make sense because of the increase in bitterness.

18.   Line 152 – in line 139 you mentioned no changes in aftertaste

19.   Table 2/3 – why the tables for each cultivar? Why not for each storage condition? That way – you could compare the cultivars for each condition. Explain in the footnote why there are no values for Q and Z for the last 7 CATA terms.

20.   Line 165 – Overall acceptability data is assumed to be continuous data for analysis purposes, wile CATA data is frequency data that is discrete in nature. Explain.

21.   Figure 4 – use bar graph – easy to show/interpret the VIPs

22.   Line 185 – the shelf time in the stores should be covered in the introduction as well.

23.   Line 202 – the texture should be discussed more. The instrumental texture should have been determined. Even if it was not determined – you should discuss the change from crunch/juicy to soft and elastic during storage.

24.   Line 210 – does the glucosinolate content go up with storage?

25.   Lines 216-18 – vague and confusing sentence

26.   Line 220 – that is true for may other vegetables as well

27.   Line 227 – is it dirty-like or dirt-like

28.   Only 6/26 references were published after 2010 – that’s a low number.

Author Response

Reviewer 2

An interesting study with fresh broccoli – there is not enough studies on the sensory properties of fresh vegetables. There are quite a few major points that the authors must take care of before this manuscript can be published. A good reference that the authors should cite in their manuscript – Jacobsson, A., Nielsen, T. & Sjöholm, I. Effects of type of packaging material on shelf-life of fresh broccoli by means of changes in weight, colour and texture. Eur Food Res Technol 218: 157.

1.      Line 12 – in the text you have mentioned 7-14 days

The authors have changed the text to match the 7-14 day window.

2.      Line change ‘organoleptic’ to ‘sensory’ in the manuscript

The authors have made this change.

3.      Line 17 – why 42 days? That’s too long in my opinion since you say in the text that they sold by a weeks’ time after arrival. Day 7 should have been a timepoint.

In the manuscript, the authors mention the arrival of broccoli is around a week or two but did no give an approximation for its sell or disposal date. This would be dependent on the seller.  However, we chose such a long-time span since to ensure we documented degradation of the produce.

4.      Line 32 – mention the usual shelf life in the grocery stores

The authors have included this information.

5.      Why did you choose Broccoli? Give the rationale and some consumption statistics related to fresh broccoli

The authors have added justification of broccoli through both economic impact and consumption statistics.

6.      Lines 52-54 – CATA is not a hybrid rapid descriptive method. The comparison is incorrect. CATA collects frequency data and does not indicate the intensities of the descriptors. Methods such as flash profiling and napping are rapid descriptive methods. So please change the explanation here.

The authors have put this information in the manuscript. While the reviewer is correct that CATA does not gather attribute intensity information, it has been shown to be able to partially replace descriptive analysis. Furthermore, recent literature does refer to CATA as a RDM regularly.

 7.      Line 63 – this hypothesis is not correct – the hypothesis leads to your research objective – is that the case here?

We’ve updated this sentence to be more appropriate to the current study.

8.      Line 78 – change ‘minutes’ to ‘min’

The authors have made this change.

9.      Line 82 – explain 7/8 cooling

This is industry terminology and has been removed for clarity.

10.   Table 1 – why wasn’t the same consumer panel used throughout the study?

While it would have been ideal, the logistical challenges were too high.

11.   Line 105 – how can it be a 15-point line scale? The word ‘point’ refers to a category scale. That is incorrect – line scales are used for trained panelists where the scores are measure using a ruler. I know that a data collection software was used which probably shows and record the number when the cursor is placed – but still it is not the right scale for untrained consumers. You should have used a 9-point category scale. Easier to understand because it corresponds to the 9-point hedonic scale in terms of the number of categories.

The scale was actually a 15-cm line scale and the authors corrected the text to reflect this. The authors used this scale to help the participants discriminate the small differences between the samples. The authors are unaware of any confusion imparted on to the participants through the scale chosen.

12.   Line 109 – version?

RedJade software does not use traditional versions, it implements continuous changes to the platform.

13.   Line 111 – random effects and fixed effects in the ANOVA model? Post-hoc mean separation technique?

The authors have included this information.

14.   Line 118 – more explanation is required on the PLS. which stat package was used for the analyses?

The authors have included this information.

15.   Line 123 – be consistent when you write ANOVA results as F values. Check throughout the results section.

The authors have fixed this to be consistent using brackets.

16.   Line 131 – an r of 0.53 is not that high.

That is true, but the authors are only speaking to statistical significance here.

17.   Line 139 – the aftertaste results does not make sense because of the increase in bitterness.

The authors are unsure of how to address this comment. The findings are odd, but no out of l

18.   Line 152 – in line 139 you mentioned no changes in aftertaste.

This statement is referring to ‘intensity’ of the aftertaste, not the CATA data mentioned ealier.

19.   Table 2/3 – why the tables for each cultivar? Why not for each storage condition? That way – you could compare the cultivars for each condition. Explain in the footnote why there are no values for Q and Z for the last 7 CATA terms.

If the CATA counts are too low it violates assumptions made by Cochran’s Q and other categorical data analysis techniques.

20.   Line 165 – Overall acceptability data is assumed to be continuous data for analysis purposes, while CATA data is frequency data that is discrete in nature. Explain.

The reviewer is correct, but predictors are not limited to continuous data in regression techniques such as PLS.

21.   Figure 4 – use bar graph – easy to show/interpret the VIPs

VIPs themselves are not easy to interpret without seeing the corresponding coefficient. The authors believe the current figure best represents the contribution of each variable to overall liking.

22.   Line 185 – the shelf time in the stores should be covered in the introduction as well.

The authors have included this information to the introduction.

23.   Line 202 – the texture should be discussed more. The instrumental texture should have been determined. Even if it was not determined – you should discuss the change from crunch/juicy to soft and elastic during storage.

The authors have added texture to the discussion, drawing off of the previously suggested reference (Jacobsson et al.)

24.   Line 210 – does the glucosinolate content go up with storage?

The authors have included this point in the discussion.

25.   Lines 216-18 – vague and confusing sentence

The authors have removed this sentence.

26.   Line 220 – that is true for many other vegetables as well

The authors have reworded this sentence to be less specific to broccoli.

27.   Line 227 – is it dirty-like or dirt-like

It is dirt-like, and the manuscript has been updated accordingly.

28.   Only 6/26 references were published after 2010 – that’s a low number.

Several of the new references are more recent.

Round  2

Reviewer 2 Report

The authors have addressed my concerns adequately.

Author Response

The authors have made the additional changes requested by the editor.